# HyperPrism: An Adaptive Non-linear Aggregation Framework for Distributed Machine Learning over non-IID Data and Time-varying Communication Links

**Haizhou Du** [*]
Shanghai University
of Electric Power
Shanghai, China

**Yijian Chen**
Shanghai University
of Electric Power
Shanghai, China

**Ryan Yang** [*]
Massachusetts Institute
of Technology
MA, USA

**Yuchen Li**
Shanghai Jiao Tong
University
Shanghai, China

**Linghe Kong**
Shanghai Jiao Tong
University
Shanghai, China

## Abstract

While Distributed Machine Learning (DML) has been widely used to achieve decent performance, it is still challenging to take full advantage of data and devices distributed at multiple vantage points to adapt and learn; this is because the current linear aggregation paradigm cannot solve inter-model divergence caused by (1) heterogeneous learning data at different devices (*i.e.*, non-IID data) and (2) in the case of time-varying communication links, the limited ability for devices to reconcile model divergence. In this paper, we present a non-linear class aggregation framework *HyperPrism* that leverages Kolmogorov Means to conduct distributed mirror descent with the averaging occurring within the mirror descent dual space; HyperPrism selects the degree for a Weighted Power Mean (WPM), a subset of the Kolmogorov Means, each round. Moreover, HyperPrism can adaptively choose different mapping for different layers of the local model with a dedicated hyper-network per device, achieving automatic optimization of DML in high divergence settings. We perform rigorous analysis and experimental evaluations to demonstrate the effectiveness of adaptive, mirror-mapping DML. In particular, we extend the generalizability of existing related works and position them as special cases within HyperPrism. For practitioners, the strength of HyperPrism is in making feasible the possibility of distributed asynchronous training with minimal communication. Our experimental results show HyperPrism can improve the convergence speed up to 98.63% and scale well to more devices compared with the state-of-the-art, all with little additional computation overhead compared to traditional linear aggregation.

## 1 Introduction

The proliferation of edge devices, such as mobile phones, wearable devices and unmanned aerial vehicles, has resulted in an exponential growth in diverse data types (e.g., images, sound and text). Addressing this surge necessitates advanced techniques capable of accurately, quickly, and practically processing vast amounts of data through efficient and scalable algorithms. Distributed Machine Learning (DML) has gained significant traction in recent years as a strategy to minimize data transfer

---

[*]Email: duhaizhou@shiep.edu.cn

[*] These authors are the same contribution.

38th Conference on Neural Information Processing Systems (NeurIPS 2024).

and computational costs by bringing computation closer to the data sources. It has become a natural solution for scaling up machine learning while preserving data privacy in various domains (*e.g.*, large language model training [39, 33], autonomous driving [60, 37], military applications [56], web search [29, 30, 28] and recommendation [32, 27, 31]).

While existing DML attempts have made significant progress, two technical barriers remain in realistic scenarios. **First**, *data heterogeneity* is one of the most critical concerns. Specifically, in a DML system where mobile devices serve as computing nodes, the data generated is typically non-independent and identically distributed (non-IID), which means the data distribution may be unbalanced across different classes or categories. Consequently, the model may exhibit bias toward the majority class or dominant data patterns, leading to suboptimal performance on the minority class or rare patterns. **Second**, traditional DML methods usually deliver poorer performance under the *time-varying communication* condition. Specifically, devices holding critical data may unexpectedly go offline or become out of range during the training, causing changes in the communication links. These interruptions can cause information loss in the aggregation, making local models no longer interchangeable and drifting away from the global model. We can summarize these powers of deviation and drift as "*divergence forces*", which drastically slows down the convergence and significantly impacts the efficiency and effectiveness of the training process in DML.

In order to tackle the above issues, we present a novel decentralized DML framework *HyperPrism*, which utilizes mirror descent [3] and employs adaptive mapping functions to project models into a mirror space, with both aggregation and gradient steps then carried out in the mapped space. Moreover, HyperPrism leverages the concept of "weighted power means" (WPM) as the aggregation function, raising the model parameters to the power of $p$, and uses HyperNetworks [12] to adaptively adjust the power degree $p$. In summary, the main contributions are summarized as follows:

- We study the problem of *divergence forces* in decentralized DML, where we particularly focus on two technical barriers due to *data heterogeneity* (i.e., non-IID data) and *time-varying communication links*. To the best of our knowledge, it is the first work to simultaneously address the challenges of non-IID data and time-varying communication links in realistic DML scenarios.
- We propose a non-linear class aggregation DML framework HyperPrism based on Kolmogorov Means, which can also be seen as mapping models to mirror space for aggregation, and we instantiate the means with an adaptive $p$ power function to enhance the convergence speed and scalability. HyperPrism achieves superior performance in its dependence on the number of devices $m$ and the power degree $p$, upgrading from $m\sqrt{m}$ to $m\sqrt[p]{m}$. This achievement also reduces the optimality gap [2] from $\sqrt{m}$ to $\sqrt[p]{m}$.
- We conduct rigorous analysis and prove that the loss bound of HyperPrism is $O((\frac{m^{P+2}}{T})^{\frac{1}{P+1}})$. In cases where few communication epochs can occur (*i.e.*, $T \leq m$), employing a larger value of $p$ yields improvements over traditional linear aggregation. Our theoretical results are consistent with state-of-the-art bounds in distributed gradient/mirror descent and single-device mirror descent.
- We carry out comprehensive experiments to assess the performance of HyperPrism framework. The experimental results demonstrate that HyperPrism achieves a remarkable acceleration in convergence speed with improvements of up to 98.63%. Moreover, HyperPrism also shows increased scalability in settings characterized by time-varying communication.

## 2 Related Work

**Decentralized DML with Data Heterogeneous.** Numerous studies have addressed non-IID data using linear solutions, such as local fine-tuning of a global model [4, 9, 8, 10], personalization in Federated Learning (FL) as a meta-learning objective [15], knowledge distillation[61], and prototype aggregation [53]. Furthermore, Li *et al.* analytically demonstrate the limitations of FedAvg on non-IID data [26]. Li *et al.* propose a variant of FedAvg by incorporating a penalty term in the local objective function [25]. Liu *et al.* propose an algorithm that capture similarities between clients to compute personalized aggregation weights for personalized FL [36]. Recently, Aketi *et al.* [1] propose a tracking-based method to mitigate the impact of heterogeneous data distribution in DML without introducing communication overhead. However, these methods, less explored in decentralized DML, are confined by the boundaries of linear aggregation.

**DML with Time-varying Communication setting.** Many DML models and their variants have been proposed to process huge amounts of data locally over time-varying communication settings. Kovalev *et al.* propose the ADOM and ADOM+ method for decentralized optimization over time-varying networks with projected Nesterov gradient descent, respectively [18, 20]. Koloskova *et al.* introduce a framework covering local SGD updates and synchronous and pairwise gossip updates on adaptive network topology [17]. De Vos *et al.* [5] introduce Epidemic Learning in decentralized learning, leading to faster convergence and improved performance by randomly changing topologies. Nedic *et al.* [44, 42] tackle the DML with topology dynamicity from the consensus perspective. They propose a model aggregation method for agents in a time-varying network topology to collaboratively solve a convex objective function, with a convergence guarantee. Moreover, future generation DML systems [20, 19, 7] also consider these time-varying communication settings as an important research area, where the naturally time-varying connectivity among pairs of mobile devices or mobile devices and edge servers will be dictated by their physical proximity.

*In this work, we focus on solving two barriers due to data heterogeneity (i.e., non-IID data) and time-varying communication links and conduct rigorous analysis to show that prior methods suffer from lower accuracy, higher loss, and slower convergence speed over time-varying networks than under a fixed topology. Additionally, their methods use linear averages for aggregation, which our framework encompasses as a particular case.*

## 3 Problem Formulation and Preliminaries

### 3.1 Problem Formulation

We consider decentralized DML with $m$ devices over time-varying communication links represented by a directed graph $G(t) = (V, \varepsilon(A^{(t)}))$, where $V$ denote vertices and $A^{(t)} = [a_{11}^{(t)}, a_{12}^{(t)}, ..., a_{mm}^{(t)}]$ is the weight matrix of the topology graph. Then the $\varepsilon(A^{(t)})$ denotes the set of directed time-changing edges between vertexes. In each time interval, we assume that the communication links between devices are symmetrical and restricted, causing random disconnections and reconnections. Each device $i$ has its own private dataset, denoted by $D_i$. Let $\mathbf{w} = (w_1, w_2, \ldots, w_m)$ denote the collection of all local models, where $w_i$ is the model held at device $i$. Then, Device $i$ constructs its local loss function as $f_i(w) = \mathbb{E}_{\zeta_i \sim D_i}[\mathcal{F}(w_i; \zeta_i, G(t))]$. Then, devices aim to solve the following optimization problem collaboratively:

$$\min_{\mathbf{w} \in \mathbb{R}^d} F(\mathbf{w}) = \sum_{i=1}^{m} f_i(\mathbf{w}), \tag{1}$$

without sharing local data (*i.e.*, without revealing $f_i$).

### 3.2 Motivation

In recent years, model merging has developed non-linear aggregation methods. One motivation for non-linear aggregation is the decreased variance in the original parameter space. HyperPrism is designed to combat stale gradients from diverged models with synchronization.

In existing work, synchronization is achieved with an aggregation mechanism, usually a mean or median, with clipping. Inspired by similarities between Mirror Descent [46] and Quasi-Arithmetic Means [16], also known as Kolmogorov Means, of the form $f^{-1}\left(\frac{1}{n} \sum_{k=1}^{n} f(x_k)\right)$, HyperPrism maps models to the dual domain before averaging to better align with the geometry of the objective function. In implementation, we focus on the special case of $\phi(w) = \frac{1}{p+1}\|w\|^{p+1}$, transforming models as $w \to w^p$, where HyperPrism therefore replaces linear means with the following weighted power mean:

$$\overline{w}_i^{(t)} = \Big(\sum_{j=1}^{m} a_{ij}^{(t)} (w_j^{(t)})^p\Big)^{\frac{1}{p}}, \tag{2}$$

where $a_{ij}^{(t)}$ is the aggregation weights of device $j$ in device $i$ at $t$ round. The WPM is a special case of the general strategy of averaging in a mirror descent dual space, but it is particularly useful due to its

ease of computation and convexity guarantees. In general, the aggregation step of HyperPrism is the following Equation (3):

$$\overline{w}_i^{(t)} = [\nabla\phi]^{-1}(\sum_{j=1}^{m} a_{ij}^{(t)} \nabla\phi(w_j^{(t)})). \tag{3}$$

But we choose to focus on $\phi(w) = \frac{1}{p+1}\|w\|^{p+1}$ which gives $\nabla\phi(w) = w^p$ (termwise exponentiation). This allows for simple run-time computation. For more intuition, in the $p = \infty$ limit the distributed averaging consensus problem [47] turns into an easier "distributed maximum problem", which is what allows for HyperPrism's tighter synchronization.

## 3.3 Distributed Mirror Descent

The phrase "distributed mirror descent" can have multiple interpretations. In the existing literature, it typically refers to the process of taking local mirror descent steps and then linearly aggregating [23, 48, 6, 59] In the case of HyperPrism, mirror descent is transformed into the distributed algorithm in a different manner. It still takes local mirror descent steps but also uses the aggregation function in Equation (3). This approach proves to be more effective because, under the optimal model $\mathbf{w}^*$, distributed mirror descent with Equation (3) remains stable, while the linear aggregation functions do not. In other papers' analysis (*e.g.*, [11], [58]), this inaccuracy is not obvious since the dominant terms are often related to communication costs, but in the perfect communication case, using a generalized mean is necessary for exact theoretical convergence.

## 3.4 HyperNetworks

Hypernetworks (HNs) are deep neural networks used to generate weights of other target networks [12]. HNs can learn the mapping relationship between the embedding vector and the target network, and adaptively generate the target network based on the input. Shamsian *et al.* apply HNs in federated learning to generate personalized model parameters [50]. Ma *et al.* present pFedLA [38] using HNs to generate aggregation weights of each model layer in personalized federated learning. Previous works [53, 55] found the different layer parameters of the local model have different impacts on model aggregation. For example, the locally learned feature representations are prone to over-fitting and thus cannot generalize well when each device only has insufficient data. In HyperPrism, the mapping function chosen is a form of the Bregman divergence, specifically $\nabla\phi(w) = w^p$, where $p$ represents the degree of WPM. Our experiments indicate that different functional layers of the model respond differently to the degree of $p$. Increasing $p$ has a more significant impact on accelerating convergence in linear or fully-connected layers. This observation inspires us to select the appropriate $p$ for different functional layers adaptively. Through the chain rule [38], we demonstrate that HN can effectively correlate the mirror mapping with the objective function, enabling it to discover the optimal $p$ for each device's model parameters in each round.

# 4 The Design of HyperPrism Framework

## 4.1 Overview

With the above preliminaries and motivation presented, we now give the design of HyperPrism framework. The framework assumes that each device $i$ maintains its local model denoted by $w_i$; for simplicity of specification, we use a round-based specification, in which the model holds by device $i$ after $t$ communication rounds is denoted as $w_i^{(t)}$.

The framework first chooses the aggregation weights $a$. Each $a_{ij}^{(t)}$, defined by

$$a_{ij}^{(t)} = \min\{e_{ij}^{(t)}, e_{ji}^{(t)}\}, \tag{4}$$

$$a_{ii}^{(t)} = 1 - \sum_{j \neq i} a_{ij}^{(t)}, \tag{5}$$

where $e_{ij}^{(t)} = \frac{1}{N_i^{(t)}+1}$ when $i, j$ are connected, otherwise $e_{ij}^{(t)} = 0$. $N_i^{(t)}$ is the number of neighboring devices $i$ at $t$-th round. $a_{ij}^{(t)}$ generated in this way satisfy Assumption 5.1. Note that the connectivity

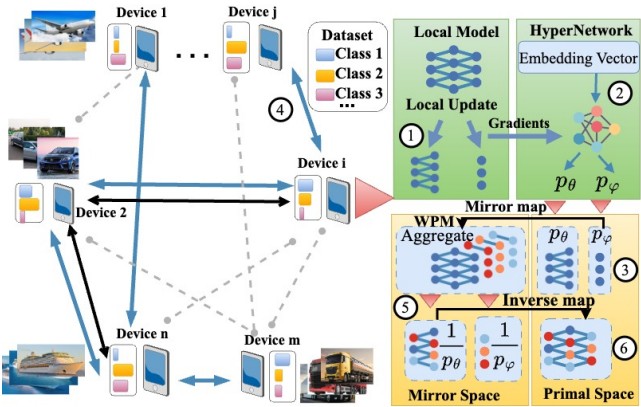

Figure 1: The Overview of HyperPrism Framework. ① Device $i$ executes local update on its own dataset. ② Device $i$ adaptively generates degrees of $p_\theta$ and $p_\varphi$ for the representation and decision parts by hypernetworks. ③ Device $i$ maps the model to the mirror space by raising the parameters with the degree of power $p_\theta$ and $p_\varphi$, respectively. ④ Device $i$ communicates with its neighbors exchanging local models. ⑤ Device $i$ aggregates received models in mirror space with WPM. ⑥ Device $i$ inverses the model to primal space by the degree of power $p_\theta$ and $p_\varphi$, then finishes the round.

$e_{ij}^{(t)}$ is determined by underlying communication systems, which consider both feasibility and security requirements (*e.g.*, low probability of detection). HyperPrism applies to generic DML, the parameter server and all-reduce frameworks can be seen as special cases (where all $a_{ij}^{(t)}$ values equal $\frac{1}{m}$).

In summary, the core of HyperPrism is that at every round $t$, device $i$ computes its local model for the next round $t+1$ as:

$$w_i^{(t+1)} = [\nabla\phi]^{-1}(\sum_{j=1}^{m} a_{ij}^{(t)} \nabla\phi(w_j^{(t)}) - \eta\nabla f_i(w_i^{(t)})), \qquad (6)$$

where $\nabla\phi$ is the mapping function with a selected degree of $p$ for each round.

## 4.2 Adaptive Degree of Power $p$ Mapping

To determine the $p$ of various layers at each round, HyperPrism establishes a relationship between the optimization problem and the degree of $p$ by utilizing hypernetworks, which adaptively choose the optimal $p$ for different layers to achieve the best performance.

Without loss of generality, HyperPrism also decouples the local model into the representation and decision parts. The representation part $\theta$ includes components like convolutional and embedding layers. The decision part $\varphi$ includes components like fully-connected layers. Thus, for device $i$, we have $w_i = \{w_{\theta,i}, w_{\varphi,i}\}$. Each device $i$ holds a local hypernetwork $HN_i$ and a randomly generated embedding vector $v_i$. Every hypernetwork consists of several fully-connected layers and employs the softmax for the output layer. $HN_i$ takes $v_i$ and gradient of the local model as input, then output $p_i = \{p_{\theta,i}, p_{\varphi,i}\}$ for $w_{\theta,i}$ and $w_{\varphi,i}$ parts, respectively. Then the mapping function evolves to $\nabla\phi(w_i) = \{(w_{\theta,i})^{p_{\theta,i}}, (w_{\varphi,i})^{p_{\varphi,i}}\}$. The hypernetwork on device $i$ can be defined as

$$p_i = HN_i(v_i; \psi_i), \qquad (7)$$

where $\psi_i$ denotes the parameters of $HN_i$. Hence, a new objective function can be derived from the original problem as

$$\min F(\mathbf{w}) = \sum_{i=1}^{m} f_i((w_i)^{HN_i(v_i;\psi_i)}). \qquad (8)$$

HyperPrism can transform the optimization problem for model parameters $w_i$ into the HN's $v_i$ and $\psi_i$. HNs adaptively output $p_{\theta,i}$ and $p_{\varphi,i}$ based on the input, and simultaneously update both $v$ and $\psi$ by gradient descent at each round. Specifically, the gradient of $v_i$ and $\psi_i$ can be computed based on the chain rule [38] as

$$\nabla_{v_i} f_i = (\nabla_{v_i}\overline{w}_i)^T \nabla_{\overline{w}_i} f_i, \qquad (9)$$

$$\nabla_{\psi_i} f_i = (\nabla_{\psi_i}\overline{w}_i)^T \nabla_{\overline{w}_i} f_i. \qquad (10)$$

---
**Algorithm 1** HyperPrism Framework.
---
1: **Input:** datasets $\{D_1, D_2, \ldots, D_m\}$, learning rate $\eta$, the number of Rounds $T$.
2: **Output:** the final model of all devices after $T$ rounds. $w^{(T)} = w_1^{(T)}, \ldots, w_i^{(T)}, \ldots, w_m^{(T)}\}$.
3: Initialize all devices' models $w^{(0)}$, hypernetworks $\psi^{(0)}$, and embedding vectors $v^{(0)}$.
4: **for** $t = 0$ to $T$ **do**
5:     **for** each device $i$ **in parallel do**
6:         Local Update: $w_i^{(t)} \leftarrow w_i^{(t)} - \eta \nabla f_i^{(t)}$.
7:         Compute $p_{\theta,i}^{(t)}$ and $p_{\varphi,i}^{(t)}$ by $HN_i(v_i^{(t)}; \psi_i^{(t)})$.
8:         Compute $\nabla \phi(w_i^{(t)}) \leftarrow \{(w_{\theta,i}^{(t)})^{p_{\theta,i}^{(t)}}, (w_{\varphi,i}^{(t)})^{p_{\varphi,i}^{(t)}}\}$.
9:         Send $\nabla \phi(w_i^{(t)})$ and $N_i^{(t)}$ to $i$'s neighbors.
10:        Receive $\nabla \phi(w_j^{(t)})$ and $N_j^{(t)}$ from neighbors.
11:        Compute weights $a_{ij}^{(t)}$ as Equation (4), (5).
12:        Aggregate received models using $a_{ij}^{(t)}$.
13:        Update $w_i^{(t+1)}$ according to Equation (6).
14:        Update $v_i^{(t+1)}, \psi_i^{(t+1)}$ as Equation (11), (12).
15:     **end for**
16: **end for**
17: **return** $w_i^{(T)}$.
---

Then, $v_i$ and $\psi_i$ can be represented as

$$v_i^{(t+1)} = v_i^{(t)} - \eta \nabla_{v_i}^{(t)} f_i^{(t)}, \tag{11}$$

$$\psi_i^{(t+1)} = \psi_i^{(t)} - \eta \nabla_{\psi_i}^{(t)} f_i^{(t)}. \tag{12}$$

Algorithm 1 demonstrates the full procedure of HyperPrism. In our real deployment, we adopt the asynchronous protocol.

## 5 Analytical Results

We rigorously analyze the properties of HyperPrism. We fully introduce all assumptions, analyze the convergence behavior of HyperPrism, and finally compare HyperPrism to previous works.

### 5.1 Assumptions

**Assumption 5.1** (Connectivity). The network graph $G(t) = (V, \varepsilon(A^{(t)}))$ and the connectivity weight matrix $A^{(t)}$ satisfy the following:

- $A^{(t)}$ is doubly stochastic for all $t \geq 1$; that is $\sum_{j=1}^{m} a_{ij}^{(t)} = 1$ and $\sum_{i=1}^{m} a_{ij}^{(t)} = 1$.

- There exists a scale $\zeta > 0$, such that $a_{ij}^{(t)} \geq \zeta$ for all $i$ and $t \geq 1$, if $\{i, j\} \in E_t$.

- There exists an integer $B \geq 1$ such that the graph $(V, E_{kB+1} \cup \cdots \cup E_{(k+1)B})$ is strongly connected for all $k \geq 0$.

**Definition 5.2** (Bregman Divergence). The Bregman Divergence of a function $\phi$ is defined as

$$D_\phi(x, y) = \phi(x) - \phi(y) - \langle \nabla \phi(y), x - y \rangle. \tag{13}$$

Note that for $\phi(x) = \|x\|^2$, $D_\phi(x, y) = \|x - y\|^2$.

**Definition 5.3** (Uniform Convexity). Consider a differentiable convex function $\phi : \mathbb{R}^d \to \mathbb{R}$, an exponent $r \geq 2$, and a constant $\sigma > 0$. Then, $\phi$ is $(\sigma, r)$-uniformly convex with respect to a $\|\cdot\|$ norm if for any $x, y \in \mathbb{R}^d$,

$$\phi(x) \geq \phi(y) + \langle \nabla \phi(y), x - y \rangle + \frac{\sigma}{r} \|x - y\|^r. \tag{14}$$

Note that for $r = 2$, this is known as *strong convexity*. This assumption also implies that $D_\phi(x, y) \geq \frac{\sigma}{r}\|x - y\|^r$.

**Assumption 5.4** (Smooth Gradient). Assume that the functions $f_i$ are convex with its gradients $\nabla f_i(\cdot)$ satisfying $L$-Lipschitz continuity [59], namely:

$$\|\nabla f_i(x) - \nabla f_i(y)\| \leq L\|x - y\|, \tag{15}$$

for all $x, y$ pairs.

This final assumption is used in nearly every mathematical analysis of DML, including [54, 3, 52, 59, 43, 51, 49]. In our analysis, it is used to bound the distance between local models.

## 5.2 Weighted Power Mean

In our experiments, we focused on the weighted power mean, generated by using $\phi(x) = \frac{1}{p+1}\|x\|^{p+1}$ which gives $\nabla\phi(x) = x^p$. Such $\phi$ is uniformly convex as seen in Proposition 5.5, and the rest of the analysis will be generalized to all uniformly convex $\phi$.

**Proposition 5.5** (Uniform Convexity of Power Functions). *For $p \geq 1$, the function $\varphi_p(x) = \frac{1}{p+1}\|x\|^{p+1}$ is uniformly convex with degree $p + 1$. This is because $\|\nabla\varphi_p(x) - \nabla\varphi_p(y)\| = \|x^p - y^p\| \geq \frac{1}{2^{p-1}}\|x - y\|$, and as a corollary we have*

$$D_{\varphi_p}(x, y) \geq \frac{1}{2^{p-1}} \cdot \frac{1}{p+1}\|x - y\|^{p+1} = \frac{1}{2^{p-1}}\varphi_p(x - y). \tag{16}$$

Given the function $\phi$, HyperPrism instructs local models to take a WPM of received models. Using the Connectivity Assumption (Assumption 5.1), we can bound the distance between local models.

**Lemma 5.6** (Consensus). *Under Assumption 5.1, for each device $i$, after $t$ rounds:*

$$\|\nabla\phi(w_i^{(t)}) - \nabla\phi(\overline{w^{(t)}})\|$$
$$\leq \vartheta(\kappa^{t-1}\sum_{j=1}^m\|\nabla\phi(w_j^{(0)})\| + \frac{m\eta G_l}{1 - \kappa} + 2\eta G_l), \tag{17}$$

*where $\vartheta = \left(1 - \frac{\zeta}{4m^2}\right)^{-2}$, $\kappa = \left(1 - \frac{\zeta}{4m^2}\right)^{\frac{1}{B}}$, $G_l = 2L\left(\max f(w_i^{(t)}) - f^*\right)$ and $\zeta$ is a constant related to the graph connectivity.*

**Theorem 5.7** (Convergence Behavior). *Consider a $(\frac{1}{2^{p-1}}, p+1)$ uniformly convex $\phi$ and the sequence $w_i^{(t)}$ under Algorithm 1 with constant step size $\eta$. Then, under Assumptions 5.1, and 5.4, if $x^*$ is the value that minimizes $F(\mathbf{w}) = \sum_{i=1}^m f_i(w)$, then*

$$\min_t[F(\overline{w}^{(t)}) - F(x^*)]$$

$$\leq \frac{4mG_l}{T}\sum_{t=0}^{T-1}\sqrt[p]{\frac{\vartheta}{2}(\kappa^{t-1}\sum_{j=1}^m\|\nabla\phi(w_j^{(t)})\| + \frac{m\eta G_l}{1 - \kappa} + 2\eta G_l)} \tag{18}$$

$$+ m \cdot \frac{p}{p+1}\sqrt[p]{2^r \cdot \eta \cdot G_l^{p+1}} + \frac{m \cdot D_\phi(x^*, \overline{w}^{(0)})}{\eta T}.$$

In this Theorem 5.7, the first term uses Lemma 5.6, and is caused by the differences between local models held at different devices. This is the effect of the *diverging forces* mentioned in earlier sections. Note that $\vartheta$ and $\kappa$ are constants related to the graph connectivity, reflecting the influence of time-varying communication links. The second term represents the error caused by a non-zero learning rate in the Distributed Mirror Descent process, and the third term represents the lingering effects of the initialization. For more proof details, please refer to the Appendix.

**Corollary 5.8.** *The bound on HyperPrism's (Algorithm 1) loss is $O(m\sqrt[p]{\eta m} + \frac{m}{\eta T})$.*

The first term becomes $O(m \cdot \sqrt[p]{m\eta})$ because since $\kappa < 1$, the $\kappa^t$ term goes to 0 as $t$ gets large. The second term is $O(m\sqrt[p]{\eta})$ is smaller than the first term, so we drop it. The final term is clearly $O(\frac{m}{\eta T})$.

**Corollary 5.9.** *For $\eta = O(\sqrt[p+1]{\frac{T^p}{m}})$, the upper bound becomes $O(m\sqrt[p]{\eta m} + \frac{m}{\eta T}) = O(\sqrt[p+1]{\frac{m^{p+2}}{T}})$.*

## 5.3 Comparison to Previous Work

Table 1: Convergence Rates in terms of $\eta$, $p$, $m$, and $T$

| Framework | $f$ error | with optimal $\eta$ | Recovered |
|---|---|---|---|
| HyperPrism | $O(m\sqrt[p]{\eta m} + \frac{m}{\eta T})$ | $O(\sqrt[P+1]{\frac{m^{P+2}}{T}})$ | NA |
| Bubeck [3] $(m=1, r=2)$ | $O(\eta + \frac{1}{\eta T})$ | $O(\frac{1}{\sqrt{T}})$ | Yes |
| Srebro [52] $(m=1)$ | $O(\eta^{\frac{1}{r-1}} + \frac{1}{\eta T})$ | $O(\frac{1}{\sqrt[r]{T}})$ | Yes |
| Nedic [44] $(GD \to r=2)$ | $O(\eta m^2 + \frac{m}{\eta T})$ | $O(\frac{m\sqrt{m}}{\sqrt{T}})$ | Yes |
| Yuan [59] $(r=2$ MD$)$ | $O(\eta m^2 + \frac{m}{\eta T})$ | $O(\frac{m\sqrt{m}}{\sqrt{T}})$ | Yes |

Table 1 summarizes the big $O$ notation convergence rates of $f(\overline{w}^{(t)}) - f^*$. Previous works also assume bounded gradients. Our analysis recovers the same bounds as the state-of-the-art in DML and single-device mirror descent with generic uniform convexity assumptions. Nedic [44] is standard gradient descent, and thus has $\phi(x) = \frac{1}{2}\|x\|^2$, which is $(1, 2)$-uniformly convex, corresponding to $p = 1$. Yuan [59] considers distributed mirror descent under strong convexity $(r = 2)$ equivalent to $p = 1$, and based on their analysis, the bound should be $O(\eta m^2 + \frac{m}{\eta T})$.

# 6 Evaluation

## 6.1 Experimental Setup

The experimental platform consists of 8 Nvidia Tesla T4 GPUs, 4 Intel XEON CPUs, and 256GB of memory. All the models and training scripts are implemented in RAY [40] and PyTorch [24].

**HyperNetworks Setup.** We construct a hypernetwork model comprising three fully-connected layers and two additional output layers activated using softmax. The outputs of the fully-connected layers are fed into each of the two output layers to generate the degree of $P$ for various parts.

**Time-varying Communication Links Setup.** We employ the NS3 platform [13] to simulate realistic time-varying communication environments consisting of multiple distributed devices. Each device is configured with the WiFi 802.11a protocol and communication among themselves in Ad-Hoc mode. To quantify the degree of connectivity of the communication links, we define the *topology density* as the ratio of available tunnels to the total tunnels.

**Models and Dataset.** We use MNIST [22] and CIFAR-10 [21] datasets distributed among devices in non-IID settings. We construct two models based on layer functionalities. The Logistic Regression (LR) [14] model consists solely of linear layers used for the MNIST dataset. On the other hand, the CNN model consists of both convolutional and fully-connected layers used for CIFAR-10.

**Non-IID Data Partitioning.** To distribute datasets in a non-IID fashion, we employ Dirichlet distribution [35] to allocate all samples among devices. The Dirichlet Degree $\alpha$ is used to control the non-IID degree. The $\alpha = 0.1$ represents the extreme scenario where each device possesses samples from only one class, while $\alpha = 10$ equals the IID scenario. These distributions reflect a challenging and realistic training environment.

**Metrics.** We consider two metrics to measure the performance of HyperPrism.

- **Average Accuracy.** We evaluate the performance of the local model per device using a global test set that contains samples with all categories. The average Top-1 accuracy of all devices is calculated in each round to measure overall performance and convergence rate.

- **Convergence Speed.** We track the loss of each round and the number of rounds to investigate the rounds needed to reach the convergence point for specific accuracy and loss.

**Baselines.** We conduct a comparative analysis of HyperPrism with state-of-the-art methods for DML in non-IID data and time-varying communication links, including $p = 1$ methods, which is one of the most influential and widely applied works in DML studies. SwarmSGD [41], Mudag [57] and ADOM [20]. To ensure a fair comparison, we made minor adjustments to each baseline method.

- *p=1*. We define it as a class of methods using a linear aggregation function, including DPSGD [34], and its variants [43, 45]. These methods can be seen as special cases of HyperPrism without the mirror mapping process.

- *SwarmSGD*. We set the number of local SGD updates equal to 1, where the selected pair of devices performs only a single local SGD update before aggregation. It can also be viewed as a special case of HyperPrism where the topology density is very low.

- *ADOM and Mudag*. We set the condition number $k$ to 10, and the number of features $d$ equals the number of classes. The gossip matrix $W(t)$ at round $t$ is chosen to be the Laplacian of time-varying communication links divided by its largest eigenvalue.

**Hyperparameters.** For all experiments, the learning rate and batch size are both fixed at $0.01$ and $128$. We generate time-varying communication graphs with different sizes and densities and evaluate 100 rounds total. The graph changes every round.

## 6.2 Experimental Results

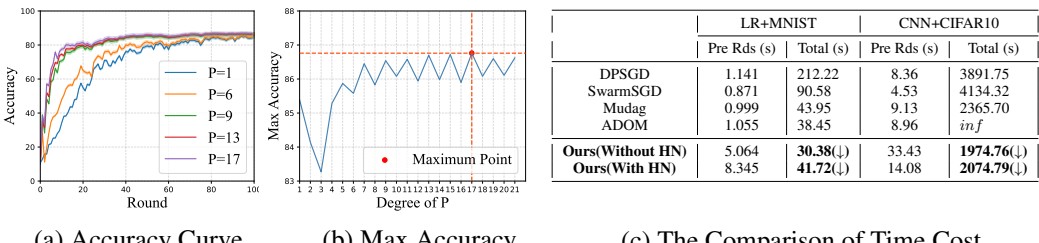

|  | LR+MNIST | | CNN+CIFAR10 | |
| --- | --- | --- | --- | --- |
|  | Pre Rds (s) | Total (s) | Pre Rds (s) | Total (s) |
| DPSGD | 1.141 | 212.22 | 8.36 | 3891.75 |
| SwarmSGD | 0.871 | 90.58 | 4.53 | 4134.32 |
| Mudag | 0.999 | 43.95 | 9.13 | 2365.70 |
| ADOM | 1.055 | 38.45 | 8.96 | $inf$ |
| **Ours(Without HN)** | 5.064 | **30.38**($\downarrow$) | 33.43 | **1974.76**($\downarrow$) |
| **Ours(With HN)** | 8.345 | **41.72**($\downarrow$) | 14.08 | **2074.79**($\downarrow$) |

|       (a) Accuracy Curve       |       (b) Max Accuracy       |       (c) The Comparison of Time Cost       |

Figure 2: The impact of different $p$ and time cost.

**Impact of different degrees of $p$.** We present the impact of varying degrees of $p$ in HyperPrism on model performance in Figure 2. In Figure 2(a), it is evident that the model converges more swiftly and attains greater accuracy as $p$ grows larger. Figure 2 (b) demonstrates that accuracy exhibits a significant fluctuation with different $p$. This underscores the substantial impact of the $p$ value selection on the performance of HyperPrism. These findings highlight the importance of choosing an appropriate value for $p$ in HyperPrism to achieve optimal performance.

**Comparison of time cost.** In HyperPrism, each hypernetwork contains only 3 linear layers with 64 nodes per layer to ensure a minimal extra computational resource cost. We record the time cost in Figure 2(c). Although HyperPrism does result in a higher time cost per iteration, it notably decreases the total number of rounds required for convergence, thereby reducing the overall time needed to achieve a specific accuracy.

**Performance of HyperPrism.** To showcase the practicality of HyperPrism, we consider the basic configuration with $Dirichlet = 0.1$, $density = 0.5$, and $m = 50$ devices. The accuracy results for all benchmarks and HyperPrism are presented in Figure 3. The convergence speed is summarized in Table 2, 3, 4. Notably, HyperPrism outperforms all benchmarks across all models. It demonstrated a remarkable superior performance over state-of-the-art baselines with convergence accuracy and convergence speed improvements of up to 4.87% and 98.63%, respectively.

**Impact of non-IID.** To investigate the impact of the non-IID Dirichlet degree on HyperPrism, we experiment with various $\alpha = 0.1, 1, 10$. The corresponding results are presented in Figure 3, and Table 2. All methods exhibit poorer performance as the non-IID degree becomes more extreme, which aligns with common intuition. However, HyperPrism demonstrates enhanced stability and faster convergence speed, especially at highly non-IID degrees.

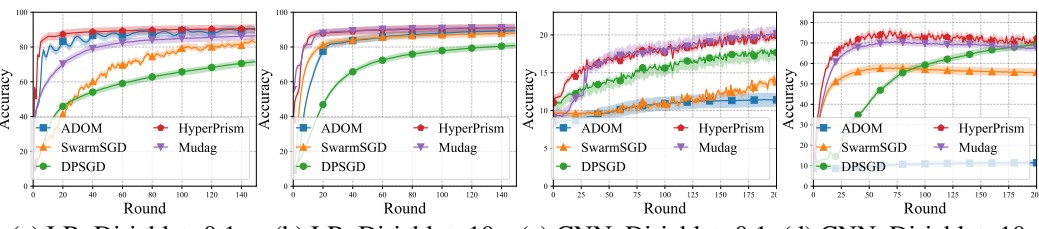

|  (a) LR, Dirichlet=0.1  |  (b) LR, Dirichlet=10  |  (c) CNN, Dirichlet=0.1  |  (d) CNN, Dirichlet=10  |

Figure 3: The Impact of non-IID Degrees

Table 2: Comparison on Different non-IID Degree

| Method | LR + MNIST | | | | | | CNN + Cifar-10 | | | | | |
|---|---|---|---|---|---|---|---|---|---|---|---|---|
| | Dirichlet = 0.1 | | Dirichlet = 1 | | Dirichlet = 10 | | Dirichlet = 0.1 | | Dirichlet = 1 | | Dirichlet = 10 | |
| | Max Acc | Conv. Rds | Max Acc | Conv. Rds | Max Acc | Conv. Rds | Max Acc | Conv. Rds | Max Acc | Conv. Rds | Max Acc | Conv. Rds |
| SwarmSGD | 83.75 | 104 | 87.01 | 47 | 83.74 | 18 | 18.24 | 198 | 51.21 | **150** | 72.22 | 14 |
| DPSGD | 71.51 | 186 | 77.60 | 173 | 71.50 | 131 | 18.04 | 187 | 43.3 | 282 | 69.2 | 92 |
| Mudag | 86.4 | 44 | 88.51 | 14 | 86.41 | **7** | 20.16 | **83** | 44.76 | 259 | 70.79 | 18 |
| ADOM | 90.58 | 11 | **90.88** | 10 | 90.58 | 23 | 10.57 | $inf$ | 11.46 | $inf$ | 12.32 | $inf$ |
| Ours | **90.61** | **6** | 90.30 | **7** | **90.60** | 8 | **20.45** | 86 | **53.26** | 169 | **75.74** | **13** |
| | (↑ **26.70%**) | (↓ **96.77%**) | (↑ **16.36%**) | (↓ **95.95%**) | (↑ **26.69%**) | (↓ **93.89%**) | (↑ **13.35%**) | (↓ **56.50%**) | (↑ **20.69%**) | (↓ **40.07%**) | (↑ **9.45%**) | (↓ **85.86%**) |

Table 3: Comparison on Different Connection Densities

| Method | LR + MNIST | | | | | | CNN + Cifar-10 | | | | | |
|---|---|---|---|---|---|---|---|---|---|---|---|---|
| | Density = 0.2 | | Density = 0.5 | | Density = 0.8 | | Density = 0.2 | | Density = 0.5 | | Density = 0.8 | |
| | Max ACC | Conv. Rds | Max ACC | Conv. Rds | Max ACC | Conv. Rds | Max ACC | Conv. Rds | Max ACC | Conv. Rds | Max ACC | Conv. Rds |
| SwarmSGD | 82.58 | 102 | 83.75 | 104 | 82.79 | 114 | 17.57 | 212 | 18.24 | 198 | 17.5 | 201 |
| DPSGD | 71.37 | 239 | 71.51 | 246 | 71.65 | 252 | 16.22 | 364 | 18.04 | 187 | 19.66 | 110 |
| Mudag | 86.39 | 44 | 86.4 | 45 | 86.36 | 45 | 20.16 | **69** | 20.16 | 86 | 20.17 | 66 |
| ADOM | 90.1 | 15 | 90.58 | 11 | 90.87 | 55 | 10.11 | $inf$ | 11.26 | $inf$ | 11.71 | $inf$ |
| Ours | **90.36** | **6** | **90.61** | **5** | **90.64** | **5** | 20.17 | 107 | **20.45** | **83** | **21.28** | **64** |
| | (↑ **26.60%**) | (↓ **97.48%**) | (↑ **26.76%**) | (↓ **97.96%**) | (↑ **26.52%**) | (↓ **98.01%**) | (↑ **24.35%**) | (↓ **70.60%**) | (↑ **13.36%**) | (↓ **50.08%**) | (↑ **21.6%**) | (↓ **68.15%**) |

Table 4: Comparison on Different Device Numbers

| Method | LR + MNIST | | | | | | CNN + Cifar-10 | | | | | |
|---|---|---|---|---|---|---|---|---|---|---|---|---|
| | World-Size = 20 | | World-Size = 50 | | World-Size = 100 | | World-Size = 20 | | World-Size = 50 | | World-Size = 100 | |
| | Max ACC | Conv. Rds | Max ACC | Conv. Rds | Max ACC | Conv. Rds | Max ACC | Conv. Rds | Max ACC | Conv. Rds | Max ACC | Conv. Rds |
| SwarmSGD | 84.74 | 93 | 83.75 | 104 | 81.54 | 116 | 22.58 | 132 | 18.24 | 198 | 14.66 | 201 |
| DPSGD | 80.65 | 137 | 71.51 | 246 | 71.57 | 372 | 22.66 | 160 | 18.04 | 187 | 17.78 | 221 |
| Mudag | 88.68 | 10 | 86.4 | 44 | 84.13 | 81 | 30.68 | **31** | 20.16 | 86 | 11.46 | $inf$ |
| ADOM | 89.01 | 20 | 90.58 | 11 | 90.73 | 42 | 11.26 | $inf$ | 10.11 | $inf$ | 9.17 | $inf$ |
| Ours | **90.01** | **7** | **90.61** | **5** | **90.75** | **3** | **31.35** | 31 | **20.45** | **83** | **19.32** | 151 |
| | (↑ **11.16%**) | (↓ **105.10%**) | (↑ **26.70%**) | (↓ **97.90%**) | (↑ **26.79%**) | (↓ **99.19%**) | (↑ **38.83%**) | (↓ **80.62%**) | (↑ **13.35%**) | (↓ **58.08%**) | (↑ **31.78%**) | (↓ **31.67%**) |

**Scalability.** We further evaluate the performance of HyperPrism with varying scales $m \in \{20, 50, 100\}$. The results are summarized in Table 4. It can be noticed that most of the baseline methods exhibit deteriorating performance as the number of devices increases. The ADOM even barely converges under 100 devices. In contrast, HyperPrism is minimally affected by the scale and maintains superior acceleration and model performance.

**Communication Graph Densities.** To further analyze the impact of connected densities on model performance, we present the performance and convergence speed of HyperPrism with various $density \in \{0.2, 0.5, 0.8\}$ in Table 3. In the extreme non-IID case, the performance of baselines deteriorates as the communication becomes denser. Particularly, ADOM exhibits significant fluctuations at a density of 0.8. However, HyperPrism maintains better performance across different densities. This can be attributed to the fact that as communication becomes denser, the information exchanged between devices becomes more complex. Given HyperPrism's resilience to non-IID scenarios, it maintains good performance in such cases.

# 7  Conclusion

In this work, we studied the important problem of *divergence forces* in decentralized DML, due to *data heterogeneity* (i.e., non-IID data) and *time-varying communication links*. We propose a non-linear class aggregation DML framework with adaptive Kolmogorov Means for aggregation to enhance the convergence speed and scalability. HyperPrism achieves superior performance in its dependence on the number of devices $m$, improving from $m\sqrt{m}$ to $m\sqrt[p]{m}$, and achieving optimality in the limit $p \to \infty$. We also conduct rigorous analysis and demonstrate that the loss bound of HyperPrism is $O((\frac{m^{P+2}}{T})^{\frac{1}{P+1}})$. In cases with few communication epochs (i.e., $T \le m$), employing a larger value of $p$ yields improvements over traditional linear aggregation. Our theoretical results are consistent with state-of-the-art bounds in distributed gradient/mirror descent and single-device mirror descent, all under a general uniform convexity assumption. To verify the effectiveness of HyperPrism, we carry out extensive experiments that demonstrate that HyperPrism achieves a remarkable acceleration in convergence speed with improvements of up to 98.63%. Moreover, HyperPrism shows increased scalability.

## Acknowledgments and Disclosure of Funding

We express our gratitude to Chengdong Ni for providing technical support. This work was supported in part by the Shanghai Municipal Education Commission grant Z2024-119, the NSFC grant 62141220, and the Yunnan Key Research Program grant 202402AD080004.

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

# 8 Appendix

## 8.1 Main Notations

We summarize the main notations in Table 5.

Table 5: Summary of Main Notations

| Symbol | Description |
|---|---|
| $m$ | The size of edge devices |
| $T$ | The total round |
| $t$ | The number of round |
| $\eta$ | The learning rate |
| $G(t)$ | The time-varying network topology graph consisting of $V$ and $\varepsilon$ |
| $V$ | The set of vertexes (*i.e.*, edge devices) |
| $\varepsilon(A^{(t)})$ | The set of directed time-changing edges between vertexes |
| $A^t$ | The weight matrix of the topology |
| $a_{ij}^{(t)}$ | The aggregation weights of device $j$ in device $i$ at $t$ round |
| $D_i$ | The private dataset of device $i$ |
| $\mathbf{w}$ | The set of all local models |
| $w_i$ | The model held at device $i$ |
| $f_i(w)$ | Device $i$'s local loss function |
| $F(\mathbf{w})$ | $F(\mathbf{w}) = \sum_{i=1}^{m} f_i(w)$ |
| $\overline{w}_i^{(t)}$ | The aggregation step of device $i$: $\overline{w}_i^{(t)} = [\nabla\phi]^{-1}(\sum_{j=1}^{m} a_{ij}^{(t)}\nabla\phi(w_j^{(t)}))$ |
| $\phi(w)$ | Define the function $\phi(w) = \frac{1}{p+1}\|w\|^{p+1}$ |
| $\nabla\phi(w)$ | The mapping function $\nabla\phi(w) = w^p$ |
| $p$ | The adaptive degree of the WPM |
| $e_{ij}^{(t)}$ | $e_{ij}^{(t)} = \frac{1}{N_i^{(t)}+1}$ when $i, j$ are connected, otherwise |
| $N_i^{(t)}$ | The number of devices neighboring device $i$ at $t$-th round |
| $\theta, \varphi$ | $\nabla\phi(w_i) = \{(w_{\theta,i})^{p_{\theta,i}}, (w_{\varphi,i})^{p_{\varphi,i}}\}$ |
| $HN_i(v_i;\psi_i)$ | The Hypernetwork on device $i$ |
| $\psi_i$ | The parameter of $HN_i$ |
| $\zeta$ | A scale related to connectivity |
| $L$ | Smoothness of gradient $\nabla f$ |
| $G_l$ | Upper bound on gradient derived from $L$ |
| $D_\phi(x, y)$ | The Bregman Divergence of a function $\phi$ |
| $\sigma, r$ | $(\sigma, r)$-uniformly convex |
| $\mathbf{y}_i^{(t)}$ | $\nabla\phi(\mathbf{y}_i^{(t)}) = \sum_{j=1}^{m} a_{i,j}^{(t)}\nabla\phi(w_j^{(t)})$ |
| $\overline{w}^{(t)}$ | $\overline{w}^{(t)} = [\nabla\phi]^{-1}(\frac{1}{m}\sum_{i=1}^{m}\nabla\phi(w_i^{(t)}))$ |
| $\vartheta, \kappa$ | Constants related to graph connectivity, containing $\zeta$ |
| $\Phi(t, s)$ | A transition matrix $\Phi(t, s) = A(t)A(t-1)\cdots A(s+1)A(s)$ |
| $\tau, k$ | round number |

## 8.2 Impact of WPM to parameters

We illustrate how HyperPrism leverages weighted power mean (WPM) to facilitate more efficient aggregation in Figure 4. By examining the distribution of model parameters, we observe that the traditional linear averaging model (w/o WPM) has approximately 11.65% of parameters lying within the range of [-0.01, 0.01] after 80 rounds. In contrast, when utilizing WPM with degrees of $p = 9$ and $p = 15$, only 1.67% and 1.21% of parameters, respectively, fall within the same range. This indicates that WPM enables the model to effectively preserve a broader range of features. Consequently, HyperPrism can extract more information from the model parameters during aggregation, leading to enhanced performance. These results underscore the effectiveness of WPM in enabling HyperPrism to capture a wider range of features and facilitate a more informative aggregation.

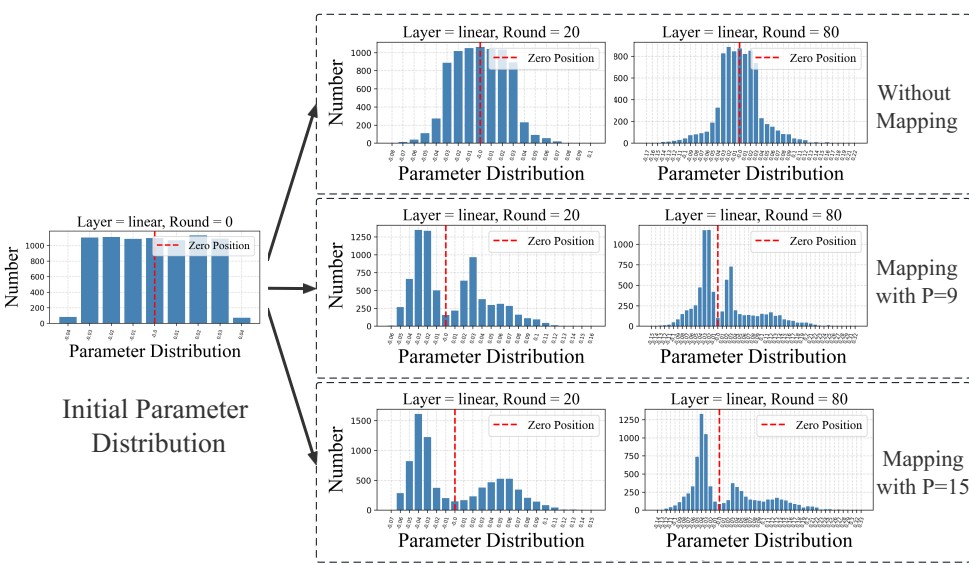

Figure 4: WPM's impact on parameter distribution.

## 8.3 Proof under Uniform Convexity

To aid analysis, introduce the sequence $\mathbf{y}_i^{(t)}$. Note that the $w_i^{(t)}$ and $\mathbf{y}_i^{(t)}$ sequences satisfy

$$\nabla\phi(\mathbf{y}_i^{(t)}) = \sum_{j=1}^{m} a_{i,j}^{(t)} \nabla\phi(w_j^{(t)}), \tag{19}$$

$$\nabla\phi(w_i^{(t+1)}) = \nabla\phi(\mathbf{y}_i^{(t)}) - \eta\nabla f_i(w_i^{(t)}). \tag{20}$$

Additionally, we can define

$$\overline{w}^{(t)} = [\nabla\phi]^{-1}(\frac{1}{m}\sum_{i=1}^{m}\nabla\phi(w_i^{(t)})) = [\nabla\phi]^{-1}(\frac{1}{m}\sum_{i=1}^{m}\nabla\phi(\mathbf{y}_i^{(t)})). \tag{21}$$

## 8.4 Proof of Lemma 5.6

*Proof.* Define a transition matrix $\Phi(t,s) = A(t)A(t-1)\cdots A(s+1)A(s)$. Then, under Assumption 5.1, Corollary 1 in [43] states that

$$\left|[\Phi(t,\tau)]_{ij} - \frac{1}{m}\right| \le \vartheta\kappa^{t-\tau}, \tag{22}$$

where $\vartheta$ and $\kappa$ are defined in the lemma statement. $\square$

We are able to write out a general formula for $\nabla\phi(w_i^{(t+1)})$:

$$\begin{aligned}
\nabla\phi(w_i^{(t+1)}) &= \sum_{j=1}^{m}[\Phi(t,k)]_{ij}\nabla\phi(w_j^{(k)}) \\
&- \eta\sum_{\tau=k+1}^{t}\sum_{j=1}^{m}[\Phi(t,k)]_{ij}\cdot\nabla f_j(w_j^{(\tau-1)}) - \eta\nabla f_i(w_i^{(t)}).
\end{aligned} \tag{23}$$

as well as $\nabla\phi(\overline{w}^{(t)})$:

$$\begin{aligned}
\nabla\phi(\overline{w}^{(t)}) &= \frac{1}{m}\sum_{i=1}^{m}\nabla\phi(w_i^{(t)}) = \sum_{j=1}^{m}\frac{1}{m}\nabla\phi(w_j^{(k)}) \\
&- \eta\sum_{\tau=k+1}^{t}\sum_{j=1}^{m}\frac{1}{m}\cdot\nabla f_j(w_j^{(\tau-1)}) - \frac{\eta}{m}\sum_{i=1}^{m}\nabla f_i(w_i^{(t)}).
\end{aligned} \tag{24}$$

It is known that for a $L$-Lipschitz continuous function $f$, if $x^*$ is the optimum of $f$, then:

$$\|\nabla f(x) - \nabla f(x^*)\| \leq 2L(f(x) - f^*)$$

Then, $\nabla \phi(w_i^{(t)}) - \nabla \phi(\overline{w}^{(t)})$ can be bounded by applying the Triangle Inequality and Equation (22):

$$\|\nabla \phi(w_i^{(t+1)}) - \nabla \phi(\overline{w}^{(t+1)})\| \leq \sum_{j=1}^{m} \vartheta \kappa^{(t-k)} \|\nabla \phi(w_j^{(k)})\|$$

$$+ \sum_{\tau=k+1}^{t} \sum_{j=1}^{m} \vartheta \kappa^{(t-\tau)} \|\eta \nabla f_j(w_j^{(\tau-1)})\| + 2\eta G_l. \tag{25}$$

Then, plugging in $k = 0$ gives

$$\leq \vartheta(\kappa^{(t)} \sum_{j=1}^{m} \|\nabla \phi(w_j^{(0)})\| + m\eta G_l \sum_{\tau=1}^{t} \kappa^{(t-\tau)} + 2\eta G_l)$$

$$\leq \vartheta(\kappa^{(t)} \sum_{j=1}^{m} \|\nabla \phi(w_j^{(0)})\| + m\eta G_l \cdot \frac{1}{1-\kappa} + 2\eta G_l). \tag{26}$$

Finally, shifting $t$ down by 1 gives the desired bound.

## 8.5 Proof of Theorem 5.7

Firstly, note that under Algorithm 1 and Assumption 5.4, Weighted AM-GM gives:

$$(r-1) \cdot \sqrt[r-1]{\frac{1}{\sigma r^{r-1} m} \cdot (m\eta G_l)^r} + m \cdot \frac{\sigma}{r} \|\overline{y}^{(t)} - \overline{y}^{(t+1)}\|^r$$

$$\geq r \cdot \left( \left( \sqrt[r-1]{\frac{1}{\sigma r^{r-1} m} \cdot (m\eta G_l)^r} \right)^{r-1} \cdot (m \cdot \frac{\sigma}{r} \|\overline{y}^{(t)} - \overline{y}^{(t+1)}\|^r)^1 \right)^{\frac{1}{r}}$$

$$= r \cdot \left( \frac{1}{r^r} \cdot (m\eta G_l)^r \cdot \|\overline{y}^{(t)} - \overline{y}^{(t+1)}\|^r \right)^{\frac{1}{r}}$$

$$= (m\eta G_l) \cdot \|\overline{y}^{(t)} - \overline{y}^{(t+1)}\|.$$

We may follow the main line of reasoning that proves Theorem 5.7.

## 8.6 Main Line of Reasoning

*Proof.* We prove bounds for generic $x$. Note that $\overline{w}^{(t)} = h^{-1}(\frac{1}{m} \sum_{i=1}^{m} h(w_i^{(t)}))$. Then, we get:

$$\eta \sum_{i=1}^{m} [f_i(w_i^{(t)}) - f_i(x)] \leq \sum_{i=1}^{m} \langle \eta \nabla f_i(w_i^{(t)}), w_i^{(t)} - x \rangle$$

$$= \sum_{i=1}^{m} \langle \eta \nabla f_i(w_i^{(t)}), w_i^{(t)} - \overline{w}^{(t)} \rangle \tag{27}$$

$$+ \sum_{i=1}^{m} \langle \eta \nabla f_i(w_i^{(t)}), \overline{w}^{(t)} - \overline{w}^{(t+1)} \rangle + \sum_{i=1}^{m} \langle \eta \nabla f_i(w_i^{(t)}), \overline{w}^{(t+1)} - x \rangle.$$

Cauchy's inequality can bound the first term. The third term can be manipulated using Equation (20). Then, combined with $f_i(\overline{w}^{(t)}) \leq f_i(w_i^{(t)}) + G_l\|\overline{w}^{(t)} - w_i^{(t)}\|$, we get

$$
\begin{aligned}
\eta \cdot (F(\overline{w}^{(t)}) - F(x)) \leq\ & (2 \cdot \sum_{i=1}^{m} \eta G_l \cdot \|w_i^{(t)} - \overline{w}^{(t)}\|) \\
& + m\eta G_l \cdot \|\overline{w}^{(t)} - \overline{w}^{(t+1)}\| \\
& + \sum_{i=1}^{m} \langle \nabla\phi(y_i^{(t)}) - \nabla\phi(w_i^{(t+1)}), \overline{w}^{(t+1)} - x \rangle.
\end{aligned}
$$

The factor of 2 comes from the 2nd term of Equation (27) added to the error from $f_i(\overline{w}^{(t)}) \leq f_i(w_i^{(t)}) + G_l\|\overline{w}^{(t)} - w_i^{(t)}\|$. Note that the last term is equal to $m \cdot \langle \overline{w}^{(t)} - \overline{w}^{(t+1)}, \overline{w}^{(t+1)} - x \rangle$. This is also equal to $m(D_\phi(x, \overline{w}^{(t)}) - D_\phi(x, \overline{w}^{(t+1)}) - D_\phi(\overline{w}^{(t+1)}, \overline{w}^{(t)}))$ by the Triangle Inequality for Bregman Divergences. We also substitute Claim 1 to replace the second term, so the value is

$$
\begin{aligned}
\leq\ & (2 \cdot \sum_{i=1}^{m} \eta G_l \cdot \|w_i^{(t)} - \overline{w}^{(t)}\|) \\
& + \frac{r-1}{r} \sqrt[r-1]{\frac{1}{\sigma m} \cdot (m\eta G_l)^r} + m\sigma\|\overline{w}^{(t)} - \overline{w}^{(t+1)}\|^r \\
& + m(D_\phi(x, \overline{w}^{(t)}) - D_\phi(x, \overline{w}^{(t+1)}) - D_\phi(\overline{w}^{(t+1)}, \overline{w}^{(t)})).
\end{aligned}
\tag{28}
$$

But, by uniform convexity, $D_\phi(\overline{w}^{(t+1)}, \overline{w}^{(t)}) \geq \sigma\|\overline{w}^{(t+1)} - \overline{w}^{(t)}\|^r$, and thus this is also

$$
\begin{aligned}
\leq\ & (2 \cdot \sum_{i=1}^{m} \eta G_l \cdot \|w_i^{(t)} - \overline{w}^{(t)}\|) + \frac{r-1}{r} \sqrt[r-1]{\frac{1}{\sigma m} \cdot (m\eta G_l)^r} \\
& + m(D_\phi(x, \overline{w}^{(t)}) - D_\phi(x, \overline{w}^{(t+1)})).
\end{aligned}
\tag{29}
$$

Then, taking the sum over $T$ gives

$$
\begin{aligned}
\eta \sum_{t=0}^{T-1} [F(\overline{w}^{(t)}) - F(x)] \leq\ & \sum_{t=0}^{T-1} 2 \cdot (\sum_{i=1}^{m} \eta G_l \cdot \|w_i^{(t)} - \overline{w}^{(t)}\|) \\
& + T \cdot \frac{r-1}{r} \sqrt[r-1]{\frac{1}{\sigma m} \cdot (m\eta G_l)^r} \\
& + m(D_\phi(x, \overline{w}^{(0)}) - D_\phi(x, \overline{w}^{(t)})) \\
\leq\ & 2m\eta G_l \cdot \sum_{t=0}^{T-1} \sqrt[r-1]{\frac{1}{\sigma}\|\nabla\phi(w_i^{(t)}) - \nabla\phi(\overline{w}^{(t)})\|} \\
& + Tm \cdot \frac{r-1}{r} \sqrt[r-1]{\frac{1}{\sigma} \cdot (\eta G_l)^r} + m \cdot D_\phi(x, \overline{w}^{(0)}).
\end{aligned}
\tag{30}
$$

Dividing through by $\eta T$, substituting $\sigma = \frac{1}{2^{p-1}}$ and $r = p+1$, and substituting $x = x^*$ and Lemma 5.6 gives the desired result. $\qquad\square$

## 8.7 Weighted Power Mean Skew Correction

**Theorem 8.1.** *Consider positive $x_i$ and $\alpha_i$ close to $\frac{1}{m}$. Then,*

$$
\left(\sum_{i=1}^{m} \alpha_i x_i\right)^{\frac{1}{p}} \approx \left(\sum_{i=1}^{m} \frac{1}{m} x_i\right)^{\frac{1}{p}} + O\left(\frac{\max|\alpha_i - \frac{1}{m}|}{p}\right).
\tag{31}
$$

*This allows us to see the weighted power mean as a way to decrease skew; this theorem is relevant in the setting of Lemma 5.6.*

Write $\alpha_i = \frac{1}{m} + \epsilon \cdot e_i$ where all $e_i \leq \frac{1}{m}$. Hold $e_i$ constant. Then, for $\alpha_i \approx m$, it is true that $\epsilon \approx 0$. The derivative of the average with respect to $\epsilon$ is

$$\frac{\sum e_i x_i^p}{p \cdot \left(\sum \frac{1}{m} x_i^p + \epsilon \cdot \left(\sum e_i x_i^p\right)\right)^{\frac{p-1}{p}}}$$

and the derivative at $\epsilon = 0$ is $\frac{\sum e_i x_i^p}{p \cdot \left(\sum \frac{1}{m} x_i^p\right)^{\frac{p-1}{p}}}$. Thus, we have the following first-order approximation:

$$\left(\sum_{i=1}^{m} \alpha_i x_i\right)^{\frac{1}{p}} \approx \left(\sum_{i=1}^{m} \frac{1}{m} x_i\right)^{\frac{1}{p}} + \epsilon \cdot \frac{\sum e_i x_i^p}{p \cdot \left(\sum \frac{1}{m} x_i^p\right)^{\frac{p-1}{p}}} \tag{32}$$

But we have $\sum e_i x_i^p \leq \sum \frac{1}{m} x_i^p$, so

$$\left(\sum_{i=1}^{m} \alpha_i x_i\right)^{\frac{1}{p}} \approx \left(\sum_{i=1}^{m} \frac{1}{m} x_i\right)^{\frac{1}{p}} + \epsilon \cdot \frac{1}{p} \left(\sum \frac{1}{m} x_i^p\right)^{\frac{1}{p}} \tag{33}$$

The $\frac{1}{p} \left(\sum \frac{1}{m} x_i^p\right)^{\frac{1}{p}}$ term is bounded, so the error is proportional to $O(\frac{\epsilon}{p})$.

## 8.8 Linear Bound on Power Mean

As a starter, we first prove a useful lemma on two real numbers with different signs. Note that all variables used in this subsection are generic ones not tied to the HyperPrism-based mechanism.

**Lemma 8.2** (Two Numbers with Different Signs). *Given any two real numbers $x > 0, y < 0$, an odd integer $p \geq 1$, and $0 \leq \alpha \leq 1$, we have*

$$\left(\alpha x^p + (1-\alpha) y^p\right)^{1/p} \geq \frac{\alpha}{2p} x + \left(1 - \frac{\alpha}{2p}\right) y. \tag{34}$$

*Proof.* Denote $C = \frac{1}{2p}$ and rewrite the right-hand side (RHS) of Equation (34) as $C(\alpha x + (1 - \alpha)y) + (1 - C)y$. We first note that we can scale both $x, y$ by $\frac{1}{|y|}$ such that $y = -1$. As such, we can simplify our proof to only focus on the case of $y = -1$. We will use shorthand $\Delta$, and rewrite as

$$(C \cdot (\alpha x + (1 - \alpha)y) + (1 - C) \cdot y)^p$$
$$= (C\alpha(x + 1) - 1)^p = -(1 - C\alpha(x + 1))^p \, \Delta^p. \tag{35}$$

We then consider two cases.

**Case 1:** $x \leq 2p - 1$. Since $-C\alpha(x + 1) \geq -C\alpha(2p) = -\alpha \geq -1$, by Bernoulli's inequality, we have

$$\Delta^p \leq -(1 + p(-C\alpha(x + 1))) = -1 + pC\alpha(x + 1). \tag{36}$$

Since $\frac{x+1}{2} \leq x^p + 1$, which can be verified by casework on $x \geq 1$ or $x \leq 1$, then

$$\Delta^p \leq -1 + \alpha(x^p + 1) = \alpha x^p + (1 - \alpha) \cdot (-1)^p. \tag{37}$$

We then finish this case by taking the power of $1/p$ on both sides.

**Case 2:** $x \geq 2p - 1$. We will denote the expression on the left-hand side and right-hand side of Equation (34) with LHS and RHS, respectively. First note that both sides of the inequality in Equation (34) are $-1$ when $\alpha = 0$ and $y = -1$. Next, we will show that the derivative with respect to $\alpha$ is always larger for the LHS of Equation (34) when $x \geq 2p - 1$.

$$\frac{d}{d\alpha} LHS = x^p + 1. \tag{38}$$

$$\frac{d}{d\alpha} RHS = pC(x + 1) \cdot (C\alpha(x + 1) - 1)^{p-1}$$
$$= \frac{1}{2}(x + 1) \cdot (C\alpha(x + 1) - 1)^{p-1}. \tag{39}$$

The LHS's derivative w.r.t $\alpha$ is clearly constant. Also, the $\frac{d}{d\alpha}RHS$ clearly has local maxima at $\alpha = 0, 1$. At $\alpha = 0$, we clearly have that since $x \geq 2p - 1$, we have

$$\frac{d}{d\alpha}LHS|_{\alpha=0} = x^p + 1$$
$$\geq \frac{1}{2}(x+1) \cdot (0-1)^{p-1} = \frac{d}{d\alpha}RHS|_{\alpha=0}.$$

(40)

For $\alpha = 1$, since $x \geq \frac{1}{2p}(x+1) - 1 \geq 0$, we have

$$\frac{d}{d\alpha}LHS|_{\alpha=1} = x^p + 1 \geq x \cdot (x)^{p-1}$$
$$\geq \left(\frac{1}{2}(x+1)\right)\left(\frac{1}{2p}(x+1) - 1\right)^{p-1}$$
$$= \frac{d}{d\alpha}RHS|_{\alpha=1}.$$

(41)

Thus, for all $0 \leq \alpha \leq 1$, $\frac{d}{d\alpha}LHS \geq \frac{d}{d\alpha}RHS$, and they are equal at $\alpha = 0$, so we always have $LHS \geq RHS$ for all $0 \leq \alpha \leq 1$ when $x \geq 2p - 1$, and we are done. $\qquad\square$

Next, consider a list of real numbers $x_i$, $i = 1, \ldots, m$. We let $M = \min x_i$ and $U = \max x_i$. We then prove the following lemma.

**Lemma 8.3.** *Assume a list of non-negative real numbers $x_i$, $i = 1, \ldots, m$ and $p \geq 1$. Given $\alpha_i$, $i = 1, \ldots, m$ such that $\sum_{k=1}^m \alpha_k = 1$ and $\alpha_i \geq 0$, we have*

$$\sum_{i=1}^m \alpha_i x_i \leq \left(\sum_{i=1}^m \alpha_i x_i^p\right)^{1/p} \leq \frac{1}{p}\sum_{i=1}^m \alpha_i x_i + \frac{p-1}{p}U.$$

(42)

*Proof.* The lower bound in Equation (42) is a direct result from the power mean inequality. As for the upper bound, we have

$$\left(\sum_{i=1}^m \alpha_i x_i^p\right)^{1/p} \leq \left(\sum_{i=1}^m \alpha_i U^{p-1} x_i\right)^{1/p}$$
$$= \left(U^{p-1}\sum_{i=1}^m \alpha_i x_i\right)^{1/p} \leq \frac{1}{p}\sum_{i=1}^m \alpha_i x_i + \frac{p-1}{p}U,$$

(43)

where the last step uses the generalized AM-GM inequality. $\qquad\square$

We now tie these two Lemmas together to prove Theorem 8.4, which to the best of our knowledge is a novel inequality on weighted power mean.

**Theorem 8.4** (Linear Bound on Weighted Power Mean). *Assume a list of real numbers $x_i$, $i = 1, \ldots, m$ and $\alpha_i$, $i = 1, \ldots, m$ such that $\sum_{k=1}^m \alpha_k = 1$ and $\alpha_i \geq 0$. For any odd integer $p \geq 1$, we have:*

$$\left(\sum_{i=1}^m \alpha_i x_i^p\right)^{\frac{1}{p}} \geq \frac{1}{2p}\sum_{i=1}^m \alpha_i x_i + (1 - \frac{1}{2p})M.$$

(44)

*Proof.* First, note that if all $x_i$s are positive, Equation (44) holds from the lower bound of Lemma 8.2. Second, if all $x_i$s are $\leq 0$, Equation (44) holds from the upper bound of Lemma 8.3 after flipping all the signs.

For the case where $x_i$s have mixed signs, let $\alpha_{pos}, \alpha_{neg}$ be the sum of the corresponding $\alpha$s for the negative and positive elements in $\mathbf{x}$, respectively. We have,

$$\left(\sum_{i=1}^{n} \alpha_i x_i^p\right)^{\frac{1}{p}} = \left(\sum_{pos} \alpha_i x_i^p + \sum_{neg} \alpha_i x_i^p\right)^{\frac{1}{p}}$$

$$\geq \left(\alpha_{pos}\left(\sum_{pos} \frac{\alpha_i}{\alpha_{pos}} x_i\right)^p + \alpha_{neg}\left(\frac{1}{p}\sum_{neg}\frac{\alpha_i}{\alpha_{neg}}x_i + \frac{p-1}{p}M\right)^p\right)^{\frac{1}{p}}$$

$$\geq \frac{\alpha_{pos}}{2p}\left(\sum_{pos}\frac{\alpha_i}{\alpha_{pos}}x_i\right) + \left(1 - \frac{\alpha_{pos}}{2p}\right)\left(\frac{1}{p}\sum_{neg}\frac{\alpha_i}{\alpha_{neg}}x_i + \frac{p-1}{p}M\right) \quad (45)$$

$$= \sum_{pos}\left(\frac{1}{2p}\right)\alpha_i x_i + \sum_{neg}\left(\frac{1}{2p^2} + \frac{2p-1}{2p^2\alpha_{neg}}\right)\alpha_i x_i$$

$$+ \left(\frac{p-1}{2p^2}\alpha_{neg} + \frac{(2p-1)(p-1)}{2p^2}\right)M,$$

where the first inequality comes from the $n$-variable same sign case in Lemma 8.3, and the second inequality comes from the 2-variable case in Lemma 8.2. Here, each positive $x_i$ is weighted with at least $\frac{1}{2p}\alpha_i$ and each negative $x_i$ is weighted with at least $\left(\frac{1}{2p^2} + \frac{2p-1}{2p^2}\right)\alpha_i$, which equals to $\frac{1}{p}\alpha_i \geq \frac{1}{2p}\alpha_i$, since $\alpha_{neg} \leq 1$.

Since $\alpha_i \geq M$ ($M < 0$ when $x_i$s have different signs), we can turn the extra $\alpha_i$ terms into $M$:

$$\left(\sum_{i=1}^{m} \alpha_i x_i^p\right)^{\frac{1}{p}} \geq \frac{1}{2p}\sum_{i=1}^{n}\alpha_i x_i + \frac{2p-1}{2p}M, \quad (46)$$

which completes our proof. $\qquad\square$

In addition to a new lower bound of HyperPrism, we can also prove an upper bound of the HyperPrism at $\frac{1}{2p}\sum_{i=1}^{m}\alpha_i x_i + \frac{2p-1}{2p}U$ following similar steps. Note that these bounds are generic and can be applied outside of DML. These bounds are necessary for analyzing the benefits of using a HyperPrism aggregation function in DML because they guarantee that the HyperPrism takes at least some minimum consideration of each $x_i$ term. This consideration is crucial because each device's local objective function is unique, and they must all be considered to optimize the overall objective function.

