# OpenReview forum: "HyperPrism: An Adaptive Non-linear Aggregation Framework for Distributed Machine Learning over Non-IID Data and Time-varying Communication Links"
_NeurIPS.cc/2024/Conference — NeurIPS 2024 poster_

### Official Review · Reviewer_uEh3 · 2024-07-11

**Soundness:** 2
**Presentation:** 3
**Contribution:** 2
**Rating:** 4
**Confidence:** 4

**Summary:**

Traditional DML methods are limited to (1) data heterogeneity and (2) time-varying communication links. In this work, they present a non-linear class aggregation framework HyperPrism that leverages Kolmogorov Means to conduct distributed mirror descent with the averaging occuring within the mirror descent dual space. The proposed method can improve the convergence speed up to 98.63% and scale well to more devices compared with the state-of-the-art, all with little additional computation overhead compared to traditional linear aggregation.

**Strengths:**

1.	The speedup of convergence is satisfactory.
2.	The theoretical discussion is sufficient.

**Weaknesses:**

1.	The authors claim that they use hypernetworks to predict p (since they use softmax, type might be int), the exponent of the mapping function. There is a little deviation from the common practice of hypernetworks, where they usually output more complex parameter weights, like the weight of classifiers (a 768 * 100 matrix, type is float). I would prefer the authors remove the part of hypernetworks, and simply introduce HN as a simple MLP.
2.	While HNs act as a crucial part of HyperPrism, the analyses of them seem limited. Since HN has softmax layers, I would be curious about (1) the value set of p and its influence, (2) the variance of p on each distributed machines when given different gradients, (3) the results of preset p (≠ 1) (like directly use the most outputted p of HNs rather than 1). I would suggest these ablation studies be added.
3.	While the performance of HyperPrism is excellent on the major experiments, the baselines used for comparison seems out of date (before 2021), it would be more reasonable if the authors could provide more recent baselines (in 2022 or 2023).
4.	The results of the experiments are a bit confusing, especially in terms of improvement. For example, in the last column of Table 1, Conv Rds of the proposed method is 13, while the best baseline is 14, how is 85.86% computed ? It is computed with the least performed baseline? It would be better if the authors could provide detailed explanations.

**Questions:**

See weaknesses

**Limitations:**

N.A

---

> ### Author Rebuttal · Authors · 2024-08-07
>
> R3-1: The difference between Hypernetwork and MLP.
>
> Response: You make an excellent point about HN and MLP. The HN itself is basically a simple MLP. However, in terms of parameter selection, there are fundamental differences between the two. First, the MLP learns a direct mapping from the determined input to the target values, treating the selection process as a prediction task. While HN models the parameter selection process as a learning problem. It is trained as a meta-model to predict the optimal parameters, based on an embedding representation of the local models. Second, the MLP only optimizes the model itself to find the best set of model parameters. In contrast, HN allows the model and the
> model embedding to be jointly optimized. The embedding vector has also been updated to better ”represent” the local models. So regardless of whether the HN outputs integers or matrices, the optimization process is quite distinct from simply optimizing an MLP. Moreover, the HN gradients are calculated based on the gradients of the local models (as shown in Equation 9), giving HyperPrism a better ability to capture the relationships between the loss function and the optimal P. Recent works such as [R1; R2] also considered similar HN-based optimization approaches, showing the potential benefits over MLP-based parameter selection.
>
> R3-2: The influence of P.
>
> Response: Due to the space limitations, some of the ablation studies are only presented in the Appendix. In Section 8.2 (lines 437-444), we use the preset fixed P to study how various P values (${P \in \mathbb{N} \mid 1 \le P \le 21}$) impact the model performance. The experimental results clearly emphasize the importance of choosing the appropriate $P$. This insight also inspires us to jointly optimize the selection of $P$ along with the performance of the model, and adaptively select the optimal $P$ during the training process.
>
> R3-3: The baselines seem out of date.
>
> Response: Although existing baseline models may not be the most recent, they are still recognized as powerful and effective methods in extreme scenarios of data heterogeneity and time-varying communication links, and have been used as baselines in recent works[R3]. We choose these baselines to ensure that our method can be fairly compared with industry standards and highlight the innovativeness and advantages of our approach in utilizing non-linear aggregation to accelerate DML training. We believe these innovations can bring new
> insight into the DML domain, and we will seriously consider exploring more novel baseline models in future work.
>
> R3-4: The confusion improvement in experimental results.
>
> Response: We apologize for any confusion caused by the performance comparisons in the results. The percentage improvements reported are all compared to the D-PSGD method, which is one of the most influential and widely applied works in DML studies. It is important to note that our proposed method not only demonstrated superior performance over D-PSGD, but also outperformed more recently published methods like ADOM and Mudag with convergence accuracy and convergence speed improvements of up
> to 4.87% and 86.36%, respectively, still demonstrating significant advantage in convergence speed. This underscores the significance and contributions of our work, as it advances brand-new insights in this research area.
>
> [R1] Xiaosong Ma, Jie Zhang, Song Guo, and Wenchao Xu. Layer-wised model aggregation for personalized federated learning. In Proceedings of the IEEE/CVF conference on computer vision and pattern recognition, pages 10092–10101, 2022.
> [R2] Aviv Shamsian, Aviv Navon, Ethan Fetaya, and Gal Chechik. Personalized federated learning using hypernetworks. In International Conference on Machine Learning, pages 9489–9502. PMLR, 2021.
> [R3] Adel Nabli and Edouard Oyallon. Dadao: Decoupled accelerated decentralized asynchronous optimization. In International Conference on Machine Learning, pages 25604–25626. PMLR, 2023.

---

### Official Review · Reviewer_5NnV · 2024-07-12

**Soundness:** 4
**Presentation:** 3
**Contribution:** 3
**Rating:** 7
**Confidence:** 4

**Summary:**

This paper addresses challenges in distributed machine learning (DML) caused by non-IID data and
unstable communication links. The proposed HyperPrism framework uses mirror descent and adaptive
mapping to project models into a mirror space for better aggregation and gradient steps.
It employs adaptive weighted power means (WPM) for efficient model aggregation, significantly
improving convergence speed and scalability. The framework's analytical results show that HyperPrism
effectively handles decentralized DML challenges, offering a robust solution for edge device data processing.

**Strengths:**

a) The paper exhibits notable originality by addressing the dual challenges of non-IID data and time-varying
communication links in distributed machine learning (DML). The HyperPrism framework introduces a novel
combination of mirror descent and adaptive weighted power means (WPM) for effective model aggregation.

b) The quality of the research is high, demonstrated through rigorous theoretical analysis and comprehensive
experimental validation, providing strong support for the framework's claims.

c) The clarity of the paper is commendable, with well-structured explanations and supportive figures and tables,
although some dense sections could benefit from further simplification.

d) The significance of the work is substantial, as it offers a robust solution to a pressing issue in DML,
with implications for improving the efficiency and scalability of edge device data processing. The results
showing enhanced convergence speed and scalability are valuable contributions to the field.

**Weaknesses:**

While the paper is strong overall, there are a few areas that could be improved.

First, some sections are densely packed with technical details, which might be challenging for readers
not deeply familiar with the subject. Simplifying these sections or providing additional explanations
could enhance accessibility.

In addition, the paper could provide more comparative analysis with existing methods to highlight the
specific advantages and potential limitations of HyperPrism. However, due to the limited period of rebuttal,
it is just optional and not necessary.

Lastly, a more detailed discussion on the computational overhead and scalability of the proposed framework
in extremely large-scale settings would be beneficial. Addressing these weaknesses would strengthen the paper
and its contributions.

**Questions:**

How does HyperPrism perform compared to other non-linear aggregation frameworks or adaptive learning methods in similar scenarios?
Are there specific benchmarks or datasets where it excels or underperforms?

**Limitations:**

The potential limitation of the proposed method is not discussed in this manuscript. It will be of great importance if the authors can give an insight to the potential readers by comparing the pro/cons of the proposed method compared to the latest works in this topic.

---

> ### Author Rebuttal · Authors · 2024-08-07
>
> R2-1: Some sections are densely packed with technical details, which might be challenging for readers.
>
> Response: To the best of our knowledge, HyperPrism is the first non-linear aggregation DML framework that combines mirror gradient descent and hypernetwork techniques. Therefore, a comprehensive explanation of the technical and theoretical details would be necessary to help readers clearly understand our approach. We will make our best efforts to provide concise overviews and technical explanations in the revised version to increase readability.
>
> R2-2: Discussion on the computational overhead and extremely large-scale settings.
>
> Response: We have conducted some experiments to compare the computational time, as presented in Section 8.3 of the Appendix (lines 446-450). The results show that HyperPrism requires more time per training round than the baseline methods. However, it significantly reduces the total number of rounds required for convergence, thereby shortening the overall time to reach a specific accuracy target. Regarding extremely large-scale settings, we conducted experiments with device sizes of {20, 50, 100} and the results show that HyperPrism is minimally affected by the number of devices and maintains excellent acceleration and model performance. Theoretically, larger scale settings will still maintain this robustness to the growth in the number of devices. Please refer to Table 3 and Section 8.9 for more details.
>
> R2-3: How does HyperPrism perform compared to other non-linear aggregation frameworks or adaptive learning methods in similar scenarios?
>
> Response: We are the first step in this new non-linear aggregation field, meaning there is almost no similar approach available for comparison. However, the non-linear aggregation mechanism in HyperPrism is performed via a specific mapping function $\phi(w) = \frac1{p+1} \lVert w\rVert^{p+1}$. According to the theory of mirror descent, it can be replaced by any convex and smooth function. This observation opens up an interesting direction for our future work, we plan to explore the use of other unique mapping functions for nonlinear aggregation, aiming to further enhance the performance and capabilities of DML systems.

---

> > ### Comment · Reviewer_5NnV · 2024-08-12
> >
> > The responses seem reasonable, so I'll stick with my original rating.

---

> > > ### Author Response · Authors · 2024-08-14
> > > **Thank you very much for your recognition.**
> > >
> > > Thank you very much for your recognition.

---

### Official Review · Reviewer_qJvT · 2024-07-24

**Soundness:** 3
**Presentation:** 3
**Contribution:** 2
**Rating:** 5
**Confidence:** 4

**Summary:**

This paper presents HyperPrism, a novel framework for decentralized machine learning (DML) that aims to address the challenges of non-IID data and time-varying communication links. The authors propose a non-linear aggregation method based on Kolmogorov Means and adaptive mapping functions, which they argue improves convergence speed and scalability compared to traditional linear aggregation methods. The paper includes theoretical analysis and experimental results to support their claims.

**Strengths:**

•	**Novelty:** The use of non-linear aggregation with adaptive mapping functions is a novel approach to simultaneously address the challenges of non-IID data and time-varying communication in DML.

•	**Theoretical Analysis:** The paper provides a theoretical analysis of the convergence behavior of their proposed approach, which is a valuable contribution.

•	**Experimental Results:** The experimental results demonstrate somewhat promising improvements in convergence speed and scalability compared to few baseline methods.

**Weaknesses:**

•	**Experimental Setup:** The experimental setup could be expanded to include more diverse datasets, models, and settings. The current experiments are limited to MNIST and CIFAR-10 with specific model architectures and the non-IID setting include only two extreme cases ($\alpha=0.1$ and $\alpha=10$). This would help to assess the generalizability of HyperPrism's performance improvements.

•	**Clarity:** The paper could benefit from improved clarity in some sections. For example, in section 3 the authors discussed HyperPrism without first introducing it. The motivation for using mirror descent and Kolmogorov Means was unclear. The connection between these concepts and the challenges of non-IID data and time-varying communication could be made more explicit.

•	**Hyperparameter Tuning:** The paper does not provide sufficient details on how the hyperparameters for the different methods were chosen. This makes it difficult to assess the fairness of the comparison.

**Others minor issues:**

•	There seems to be a missing index $i$ on $w$ in equation 1.

•	Heterogeneity in referencing in related work section. Some references seem to be typed manually (or at least they are not connected to any entry in bibliography).

**Questions:**

See weaknesses.

**Limitations:**

Overall, the paper addresses an important problem in DML and proposes a novel solution with promising theoretical and empirical results. However, the experimental setup and clarity issues mentioned above make it just fall short of this venue’s bar.

---

> ### Author Rebuttal · Authors · 2024-08-07
>
> R1-1: Experiments are limited to MNIST and CIFAR-10 and the non-IID setting includes only two extreme cases.
>
> Response: The MNIST and CIFAR-10 are the most commonly used datasets in the DML field, and recent works such as [R1] have also chosen these datasets for verification. Moreover, due to space limitations, some experimental results have not been presented in figures, including various Non-IID settings (α = {0.1, 1, 10}), topology density, number of devices, etc. The proposed HyperPrism demonstrates superiority in both convergence speed and accuracy under various settings. Please refer to Tables 1, 2, and 3 in the experimental
> section for details (lines 281).
>
> R1-2: The motivation for using mirror descent and Kolmogorov Means.
>
> Response: Model Merging is a growing field, which in recent years has developed nonlinear aggregation methods of their own. This constitutes the ultra-low update frequency update solution, with one update post-training. For ultra-high frequency updates, i.e., once per gradient, linear aggregation is optimal. However, for this vast swathe of space between these frequencies, other solutions should emerge.  One motivation for using nonlinear aggregation is the decreased variance in the original parameter space. Our system is designed based on the thesis that combining gradients from different models is dangerous if the gradient is computed at vastly different sets of parameters. Thus, the main problem is synchronizing these sets of parameters. Kolmogorov Means let you tune this, for example, with weighted power means of $p=11$, all parameters go immediately to values near the maximum value 25 (by calculating).  Meanwhile, in DML, the primary challenges are data heterogeneity and time-varying communication links. Traditional linear aggregation struggles to address the model divergence stemming from these issues, which hurts performance. The proposed HyperPrism maps models to a dual domain to better align with the geometry of the objective function. It introduces a specific mapping function, $\phi(w) = \frac{1}{p+1} \lVert w\rVert^{p+1}$, transforming models as $w \rightarrow w^p$. Then, the Kolmogorov Means method is applied to achieve nonlinear aggregation in the form of Weighted Power Mean (WPM), enabling HyperPrism to capture a broader array of features, thus making it particularly suitable for scenarios with data heterogeneity and time-varying communication links. We illustrate how HyperPrism leverages WPM to facilitate more efficient aggregation in the Appendix; please refer to section 8.4 for details (lines 451-459).
>
> R1-3: How the hyperparameters for the different methods were chosen to ensure fairness.
>
> Response: We use the same basic hyperparameters, such as model structure, learning rate, batch size, optimizer, etc., for all baselines. For methods with unique hyperparameters, we also made targeted adjustments to ensure a fair comparison between all methods. Details are presented in the Baselines section (lines 266-275).
>
> [R1] Martijn De Vos, Sadegh Farhadkhani, Rachid Guerraoui, Anne-Marie Kermarrec, Rafael Pires, and Rishi Sharma. Epidemic learning: Boosting decentralized learning with randomized communication. Advances in Neural Information Processing Systems, 36, 2024.

---

> > ### Comment · Reviewer_qJvT · 2024-08-13
> >
> > Thank you for the detailed response. I acknowledge that I missed some of the additional settings presented in table 1. Therefore I am willing to increase my score to 5. I will not go beyond score of 5 because I still find the diversity of the experiments limited (e.g. datasets, models, lack of recent baselines).
> >
> > PS: could you also address the minor issues I mentioned in my review?

---

> ### Author Response · Authors · 2024-08-13
> **Thank you very much for your recognition!**
>
> Thank you very much for your valuable suggestions and acknowledgment. It is essential to highlight that our method is specifically designed for decentralized machine learning scenarios with strong data heterogeneity and time-varying communication networks. Excessively complex large datasets often hardly make achieving global convergence extremely challenging, even non-convergence. This is why existing studies in this field predominantly use the MNIST and CIFAER10 datasets for evaluation, e.g., [R1, R2]. We also actively explore implementing our method across a broader spectrum of scenarios.
>
> The responses to the two above minor issues are as follows:
>
> 1) We confirm no additional index 'i' is needed in Equation (1). The 'w'  of Equation (1) denotes the aggregated local model from neighbors' devices, represented as $f_i(w)=\mathbb{E}_{\zeta_i\sim D_i} [\mathcal{F}(w_i;\zeta_i, G(t))]$ (Please refer to Line 109, Page 3). The notation in Equation (1) then represents selecting a single model which optimizes the function F(w) = sum of f_i(w).
>
> 2) We confirm that our manuscript is generated using the NeurIPS 2024 LaTeX template, and manual reference typing is not applicable in this context. There may be some formatting issues during the PDF conversion process that lead to some references not being linked to the bibliography.
>
> [R1] Vogels T, He L, Koloskova A, et al. Relaysum for decentralized deep learning on heterogeneous data. Advances in Neural Information Processing Systems, 2021, 34: 28004-28015.
>
> [R2] Le Bars B, Bellet A, Tommasi M, et al. Refined convergence and topology learning for decentralized SGD with heterogeneous data. International Conference on Artificial Intelligence and Statistics. PMLR, 2023: 1672-1702.

---

### Author Rebuttal · Authors · 2024-08-07

We thank all reviewers for their careful reviews and constructive suggestions. Our responses to the main issues are below：

---

### Decision · Program_Chairs · 2024-09-25

**Decision:**

Accept (poster)

**Comment:**

This paper proposes a novel way of non-linear aggregation in distributed machine learning, with the use of a hyper network. The advantage of this method is the reduction of communication rounds especially in the presence of non-IID data and time-varying communication links, thereby improving the convergence speed. The convergence of the algorithm has been analyzed theoretically and its performance has also been validated in experiments. It would be useful if the theoretical convergence results can be discussed further, such as by comparing the convergence rate with linear aggregation algorithms and explaining whether/how the convergence bound captures the effect of non-IID data and time-varying communication links. In addition, as mentioned by the reviewers, the presentation of this paper could be improved.